# The Role of Polycomb Proteins in Cell Lineage Commitment and Embryonic Development

Chet H. Loh and Gert Jan C. Veenstra *

Department of Molecular Developmental Biology, Radboud University, 6525 GA Nijmegen, The Netherlands
* Correspondence: g.veenstra@science.ru.nl

**Abstract:** Embryonic development is a highly intricate and complex process. Different regulatory mechanisms cooperatively dictate the fate of cells as they progress from pluripotent stem cells to terminally differentiated cell types in tissues. A crucial regulator of these processes is the Polycomb Repressive Complex 2 (PRC2). By catalyzing the mono-, di-, and tri-methylation of lysine residues on histone H3 tails (H3K27me3), PRC2 compacts chromatin by cooperating with Polycomb Repressive Complex 1 (PRC1) and represses transcription of target genes. Proteomic and biochemical studies have revealed two variant complexes of PRC2, namely PRC2.1 which consists of the core proteins (EZH2, SUZ12, EED, and RBBP4/7) interacting with one of the Polycomb-like proteins (MTF2, PHF1, PHF19), and EPOP or PALI1/2, and PRC2.2 which contains JARID2 and AEBP2 proteins. MTF2 and JARID2 have been discovered to have crucial roles in directing and recruiting PRC2 to target genes for repression in embryonic stem cells (ESCs). Following these findings, recent work in the field has begun to explore the roles of different PRC2 variant complexes during different stages of embryonic development, by examining molecular phenotypes of PRC2 mutants in both in vitro (2D and 3D differentiation) and in vivo (knock-out mice) assays, analyzed with modern single-cell omics and biochemical assays. In this review, we discuss the latest findings that uncovered the roles of different PRC2 proteins during cell-fate and lineage specification and extrapolate these findings to define a developmental roadmap for different flavors of PRC2 regulation during mammalian embryonic development.

**Keywords:** Polycomb; H3K27me3; H2AK119ub; pluripotency; embryogenesis; lineage commitment

## 1. Introduction

For a single pluripotent cell to expand and differentiate into a specialized and committed cell type in the body, numerous cell-fate decisions and lineage choices must be made during its development. As cells cascade through the landscape of lineage decision steps regulated by epigenetic thresholds, correct genes need to be switched on and genes controlling alternative lineages must be repressed in a timely and dynamic manner.

Polycomb proteins play a crucial role in regulating these processes. There are two main types of Polycomb protein complexes. Both can contribute to repression of transcription of target genes, but they exhibit different histone-modifying activities. Polycomb Repressive Complex 1 (PRC1) contains E3 ligase activity that mediates ubiquitination of lysine 119 residues of histone H2A (H2AK119ub) [1], whereas Polycomb Repressive Complex 2 (PRC2) catalyzes the mono-, di-, and tri-methylation of lysine 27 on histone H3 tails (H3K27) [2,3]. The two complexes are very different in their composition—with the core catalytic component of PRC2 consisting of EZH1/2 (Enhance of Zeste 2 homolog), EED, SUZ12, and RBBP4/7 proteins [4–6] and PRC1 composing of at least eight different variants of interacting proteins around the core of RING1A/B (RING1, RNF2) and different PCGF proteins. Both complexes are associated with transcriptional silencing [7,8] and they can do so cooperatively, e.g., JARID2 from PRC2 is able to bind to PRC1 deposited H2AK119ub mark [7,9–12].

Seminal work done in mouse and Drosophila has elucidated the roles of PRC1/2 in maintaining pluripotency by repressing the expression of master regulators of differentiation in embryonic stem cells. Intriguingly, they also play a crucial role just a short while after the exit of pluripotency by maintaining the precise repression of Hox genes clusters which are crucial during the body-axis formation and segmentation [2,13,14]. The loss of catalytic subunits, e.g., deletion of EZH1/2 and EED in mouse embryos, leads to gross defects in gastrulation and embryonic lethality [15–17]. By contrast, the effect of losing variant subunits, such as MTF2 and JARID2, which interact with the catalytic core at substoichiometric ratios, varies. *Mtf2* null mice display embryonic lethality much later during development, at around E17.5, and *Jarid2* null mice are unable to survive past E14.5 [18–21]. In some cases, such as the loss of AEBP2 (PRC2.2), mouse embryos were able to survive but displayed defects in skeletal muscle formation [22,23].

In this review, we aim to dissect the dynamic roles of Polycomb proteins during development, with a main focus on what is known in mouse and human. Recent advances in in vitro 3D/2D differentiation systems allow us an unprecedented access to study specific roles of epigenetic regulators, such as PRC1/2, during different time points of development. In this review, we will integrate the recent findings on the role of Polycomb proteins in development, with evidence from in vitro models such as Embryoid Bodies (EBs), monolayer differentiation systems, and in vivo mouse embryo studies. We will discuss these findings and chart them chronologically during development, which combine to represent a developmental timeline that is, in part, regulated by different and dynamic Polycomb proteins. This overview provides further insight into the diversity and specificity of lineage specification defined by the wide repertoire of Polycomb proteins.

## 2. More Than the Sum of Its Parts—Polycomb Proteins and Their Molecular Functions

The biochemical properties of Polycomb complexes have important implications for their roles in development. Biochemically distinct variant complexes have been described for both PRC1 and PRC2 (Figure 1). PRC1 is made up of at least eight different variant forms and they are classified based on different interactors with the E3 Ligase RING1A/B (RING1, RNF2) [1]. Canonical PRC1 (cPRC1) consists of RING1A/B with SCMH and PHC proteins which are associated with chromatin compaction, PCGF2/4 proteins that ensure the stabilization and integrity of the complex, and CBX proteins which both compact chromatin and aid in targeting of PRC1 to genomic loci [24–28]. Non-canonical PRC1, on the other hand, contains either the RING1 and YY1 binding protein (RYBP) or YY1-associated factor 2 (YAF2) protein associating with RING1A/B and different variants of the PCGF proteins [29–31]. For example, the PRC1.1 variant complex comprises the ncPRC1 core consisting of RING1A/B and RYBP/YAF2, together with PCGF1 and KDM2B. KDMB2B is crucial in targeting RING1B to unmethylated CpG islands. Functionally, key differences between cPRC1 and ncPRC1 include their ability to interact with H3K27me3 via the chromodomains found in CBX proteins of cPRC1 [12], and the much lower ubiquitin ligase activity of cPRC1 compared to ncPRC1 complexes. The interaction of CBX proteins of cPRC1 with H3K27me3 underscores the potential for functional interactions with PRC2 activity [32]. ncPRC1 complexes have other mechanisms of recruitment that are independent of PRC2 and H3K27me3 [31,33].

PRC2 comprises two major variant complexes, namely PRC2.1 and PRC2.2. Both contain the catalytic core made up of EZH1/2, SUZ12, EED, and RBBP4/7 proteins [5,34,35], but the interaction partners of the core complex are different. PRC2.1 contains one of the Polycomb-like (PCL) proteins—MTF2, PHF1, or PHF19. The PCLs are not directly involved in the catalytic activity of the complex, but MTF2 has been shown to play major roles in the direct recruitment of PRC2 to Polycomb target genes at unmethylated CpG islands [36,37]. In addition to a PCL protein, PRC2.1 contains either EPOP (Elongin BC, PRC2 associated protein) or PALI1/2 (PRC2-associated ligand-dependent co-repressor isoform 1/2) [38,39]. PRC2.2, on the other hand, contains the AEBP2 and JARID2 proteins, both well studied and established accessory subunits of PRC2 [22,40]. AEBP2 has been characterized with

nucleosome binding affinity that could stimulate the catalysis of H3K27 methylation [41]. JARID2, a member of the Jumonji family of proteins that demethylates histone proteins, is known to cooperatively stimulate PRC2 catalytic activity with AEBP2 and facilitate core PRC2 binding to DNA in vitro, particularly towards narrow PRC2 binding domains [42–45].

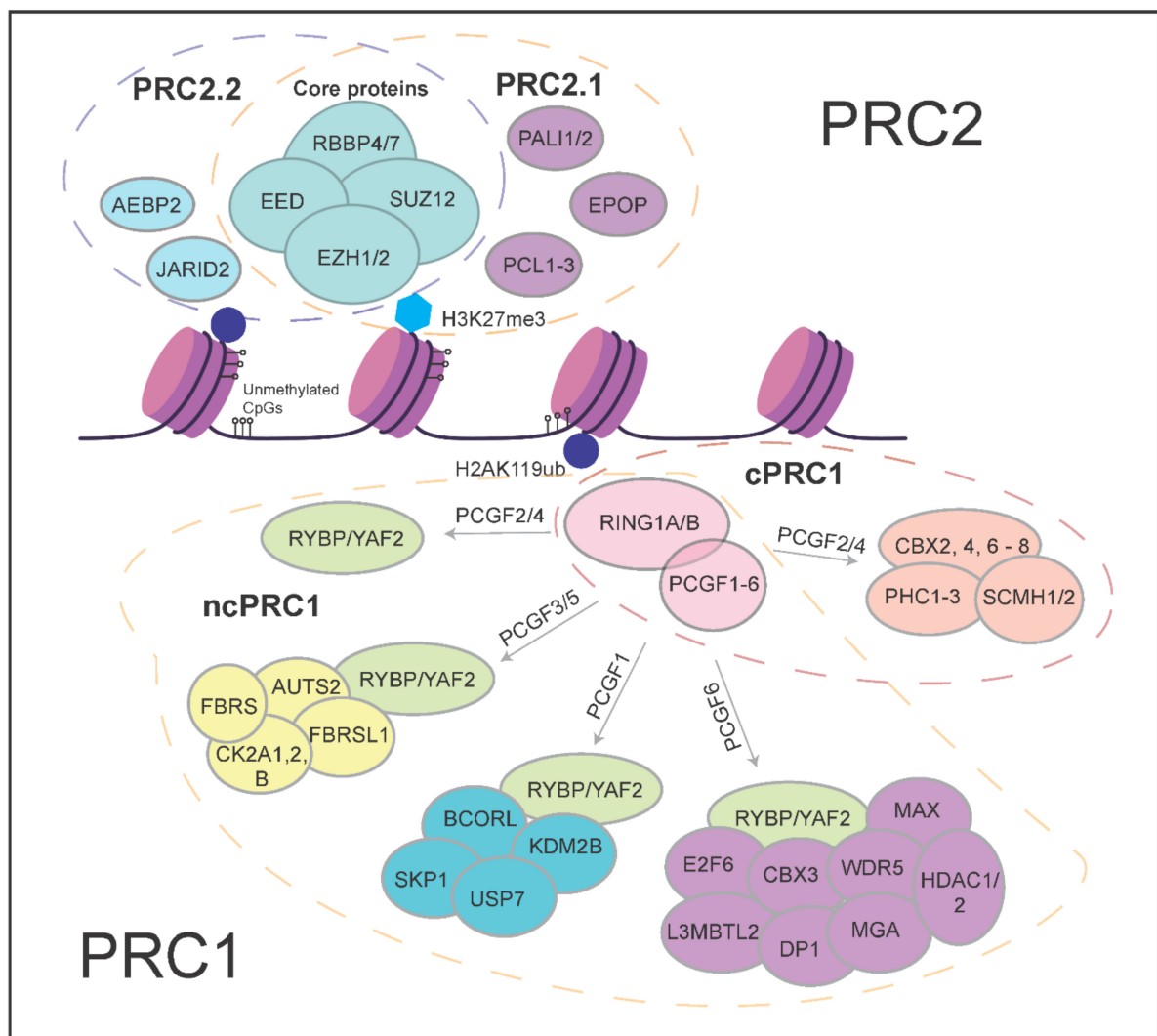

**Figure 1.** Schematic overview of the different Polycomb group proteins.

Additionally, JARID2, a part of the PRC2.2 complex, is able to bind to H2AK119ub, which is deposited by PRC1. This increases PRC2 occupancy and deposition of H3K27me3 by PRC2, constituting a positive feedback loop from PRC2 to PRC2 [7,9,11] (Figure 2). As previously discussed, the recognition and binding of H3K27me3 mark by the CBX proteins of cPRC1 complex outlines an additional layer of communication between PRC1 and PRC2 as cPRC1 can also catalyze H2AK119 ubiquitylation, albeit at lower levels compared to variant PRC1 complexes. These feedback loops (Figure 2, dotted arrows) highlight the major dependency of individual components on maintaining Polycomb repressive mechanisms, with the reduction of any one component affecting others (PRC1 binding down in PRC2 mutants and vice versa [45,46]). One further example of how PRC2 and PRC1 communicate between the complexes is demonstrated by EED, a core component of PRC2, which can bind to H3K27me3 to allosterically reinforce both PRC2.1 (MTF2-directed recruitment) and PRC2.2 (JARID2 binding to H2AK119ub) [32,46] activity.

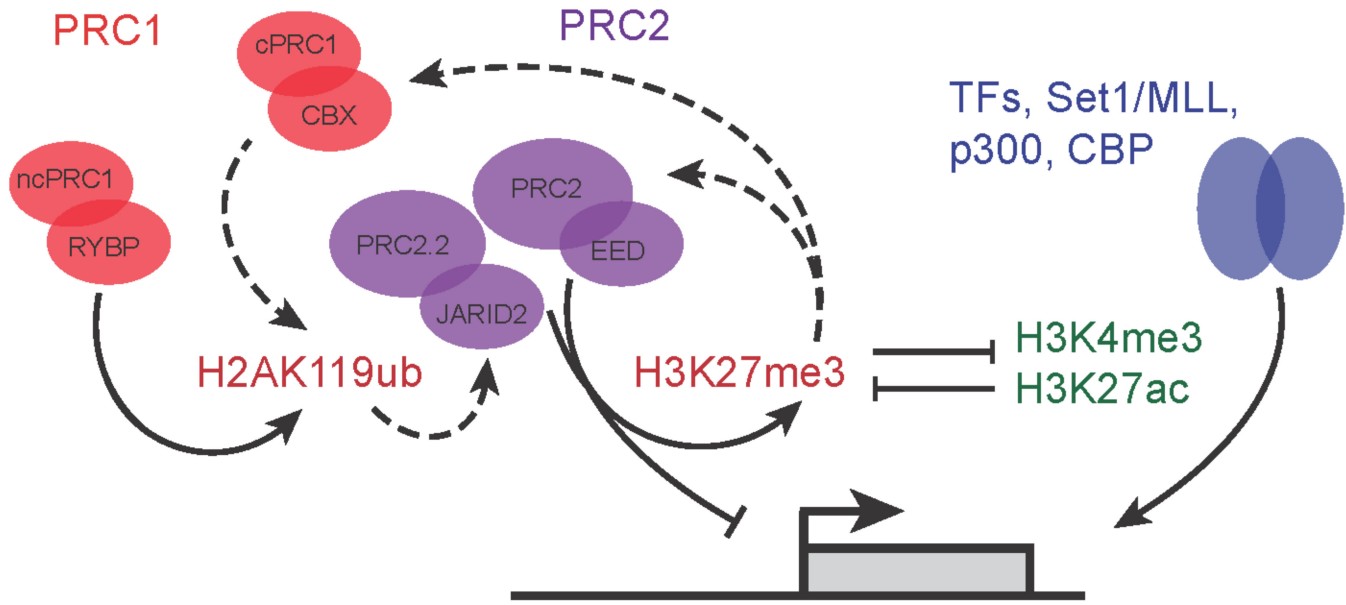

**Figure 2.** Feedback regulatory loops between activating and repressive complexes. Dashed arrow lines represent the feedback and cooperation between PRC1 and PRC2 variant complexes.

The interactions of the PRC complexes form intertwined auto-regulatory and bistable switch circuitry (Figure 2). The PRC2 subunit EED binds to H3K27me3 deposited by PRC2 subunit EZH1/2. PRC2.2 subunit JARID2 binds to H2AK119 deposited by ncPRC1 (and to a lesser extent cPRC1), whereas some of the CBX subunits of cPRC1 bind to H3K27me3 to mediate chromatin compaction. These auto-regulatory loops mediate strong interdependencies of PRC1 and PRC2 subunits. A deficit in one subunit or complex affects most if not all PRC complexes [32,36,45,46]. These auto-regulatory loops are mirrored on the side of gene activation. The permissive "Trithorax" mark H3K4me3, deposited by SET1/MLL complexes, is bound by transcription initiation factor TFIID [47], leading to recruitment of the initiating form of RNA polymerase II, which in turn can recruit SET1 [48]. While both Polycomb and "Trithorax" complexes are subject to positive autoregulatory feedback, the interactions between these repressive and activating systems is mutually antagonistic (Figure 2). H3K4me3 and H3K27me3 are mutually inhibitory, whereas the enhancer mark H3K27ac is mutually exclusive with methylation of the same lysine. The mutual antagonism mediates bistable switch behavior, and the stability of both ON and OFF states is stabilized by positive auto-regulatory behavior. This has important implications for how Polycomb contributes to developmental gene regulation. The mutually exclusive and competitive binding of repressive and activating mechanisms presents an epigenetic threshold that controls transcription of important developmental regulators [49]. Indeed, the loss of repressive Polycomb machinery allows for uncontrolled and rapid induction and binding of activating complexes to promote cell lineage-specific gene expression [8].

## 3. Roles of Polycomb Group Proteins in Pluripotency and Early Development

Based on the roles of Polycomb in homeobox gene regulation in *D. melanogaster* development [50–53], it is no surprise that the Polycomb proteins serve important roles in mammalian development. Early loss-of-function studies in mice have outlined the critical importance of core Polycomb proteins, particularly in early developmental processes such as gastrulation and body-axis formation [17,54–56]. Although the studies were conducted using various loss-of-function methods, the picture emerged that embryos could not survive past gastrulation, without the PRC2 core proteins [17,55,56]. Recent studies identified an important role for PRC2 in restricting trophoblast induction in human naïve pluripotent stem cells in vitro [57,58], suggesting a role in the earliest mammalian lineage decision after fertilization. In PRC1, the loss of RING1B protein results in developmental arrest

at E6.5, during gastrulation, but RING1A mutants were able to survive and reach birth, although displaying defects in transformation of the axial skeleton and altered levels of Hox gene expression [59,60].

On the other hand, the loss of some other non-core subunits, such as the PCLs and JARID2/AEBP2 of PRC2 in mice, revealed a later embryonic lethality [23,61]. Though these subunits are essential for normal development, this could suggest that their role in overall PRC2 activity is more subtle, is buffered by other activities at earlier stages, or is more selectively required for specific cell lineages or developmental processes. Their roles may also be contextualized by the expression levels of both PRC2 and the activators they antagonize in specific cell lineages and stages of differentiation. Table 1 provides an overview of the Polycomb loss-of-function mutations during development, outlining the different cell-types or lineages that the Polycomb proteins were studied in.

**Table 1.** In vivo and in vitro phenotypes of Polycomb mutants.

| Complex | Variant Complex | Non Core Subunit | In Vivo Developmental Phenotype | In Vitro Differentiation Phenotype | Cell Type or Tissue Specificity | References |
|---|---|---|---|---|---|---|
| PRC2 | PRC2.1 | MTF2 | Embryonic lethal due to severe anemia around embryonic day 15.5 | *Mtf2* null cells result in faster differentiation towards all cell types in EBs | ESC/Gastrula | Faust et al., 1995: Hojfeldt et al., 2018; Perino et al.,2018; Loh et al., 2021; Rothberg et al., 2018 |
| | | PHF1 | ND | Increased stoichometry relative to EED during NPC differentiation | Neuronal Precursors | Kloet et al., 2016 |
| | | PHF19 | ND | Enhanced erythrocyte differentiation | Mesodermal | García-Montolio et al., 2021 |
| | | EPOP | ND | ND | ND | |
| | | PALI1/2 | Lethal between E11.5 and perinatal lethality | Deregulation and reduction of H3K27me3 on PRC2 target genes | ND | Conway, E. et al., 2018 |
| | PRC2.2 | JARID2 | Lethal between E10.5 - E18.5 | Stagnated differentiation towards all germ layers | ND | Loh et al., 2021 |
| | | AEBP2 | Late-embryonic and perinatal lethality; skeletal transformations | Deregulation and increase of H3K27me3 on PRC2 target genes | ND | Grijzenhout, A. et al., 2016; Kim, H. et al., 2011 |

**Table 1.** *Cont.*

| Complex | Variant Complex | Non Core Subunit | In Vivo Developmental Phenotype | In Vitro Differentiation Phenotype | Cell Type or Tissue Specificity | References |
|---------|-----------------|------------------|--------------------------------|-----------------------------------|-------------------------------|------------|
| cPRC1 | | CBX2 | Male to female sex reversal | Activator of the testis - determining gene *Sry* | Testes, Sex Organ | Katoh-Fukui, Y. et al., 1998 |
| | | CBX4 | Pre-weaning lethality | ND | ND | |
| | | CBX6 | Several transformational defects in late embryonic development | ND | Enriched in NPC differentiation | Dickinson, M. E. et al., 2016 |
| | | CBX7 | ND | Enriched in early ESC stages | | |
| | | CBX8 | ND | ND | Enriched in NPC differentiation | Morey, L. et al. 2012 |
| | | PHC1 | Perinatal lethality | Destabilized pluripotency by disruption of Nanog long-range interactions | ND | Isono K. et al., 2005 |
| | | PHC2 | Posterior transformation; Hox gene deregulation; Viable birth | ND | ND | Isono K. et al., 2005 |
| | | PHC3 | Phc3 null animals are born but exhibit cardiac abnormalities | ND | ND | Dickinson, M. E. et al., 2016 |
| | | PCGF2 | Posterior transformation; Hox gene deregulation; Viable birth | ND | ND | Akasaka, T. et al., 1996 |
| | | PCGF4 | Perinatal lethality, ataxia, severe haematopoietic defects | ND | ND | van der Lugt, N. M. et al., 1994 |

**Table 1.** *Cont.*

| Complex | Variant Complex | Non Core Subunit | In Vivo Developmental Phenotype | In Vitro Differentiation Phenotype | Cell Type or Tissue Specificity | References |
|---|---|---|---|---|---|---|
| ncPRC1 | | CBX3 | Defects in germ cell development | Inhibition results in neural differentiation impairment and diverting towards mesendodermal differentiion | Neural Precursors | Abe, K. et al., 2011 |
| | | PCGF1 | E12.5 lethality | Promotes ectoderm and mesoderm differentiation | Ectoderm and Mesoderm lineages | Yan Y et al., 2017 |
| | | PCGF3 | Double Pcgf3;Pcgf5 deletion results in female-specific embryonic lethality | ND | ND | Almeida, M. et al., 2017 |
| | | PCGF5 | | | | |
| | | PCGF6 | defects in pre-implantation and periimplantation and in placenta development | Ablation of Pcgf6 in ESCs leads to robust de-repression of such germ cell-related genes, in turn affecting cell growth and viability. | pre- and peri-implantation mouse embryo | Dickinson, M. E. et al., 2016 |
| | | KDM2B | Mid-gestation lethality; posterior transformation of the axial skeleton | Maintain pluripotency by recruitment of PRC1 to CpGs | ND | Boulard, M. et al., 2015 |
| | | RYBP | E6.5 Lethality; impaired cell proliferation | Depletion of Rybp inhibits proliferation and promotes Neuronal differentiation of embryonic neural progenitor cells (eNPCs) | Neural Precursors | Pirity, M. K. et al., 2015 |
| | | AUTS2 | Die before weaning with defects in the nervous, cardiovascular and biliary systems | ND | Neural Ectoderm | Gao, Z. et al., 2014 |
| | | L3MBTL2 | E6.5 lethality; gastrulation defects | Compromised proliferation and abnormal differentiation of L3mbtl2(-/) embryonic stem (ES) cells | ESC | Qin, J. et al., 2012 |

Most of the biochemical and functional studies done in vitro were performed in mouse embryonic stem cells (mESCs). They model the early stages of development well,

particularly the inner cell mass of the blastocyst, in which Polycomb proteins play an important role in regulating pluripotency and cellular differentiation. Recent studies have focused more on the mechanisms of how non-core Polycomb proteins work in regulating gene expression in these mESCs.

### 3.1. PRC2 Variant Complexes and Their Roles in Embryonic Stem Cells

In terms of PRC2, several major landmark papers uncovered the mechanism by which MTF2 (PRC2.1) can directly and selectively bind to a subset of unmethylated CpG-dense DNA regions, with preferential binding to sites with specific DNA shape properties such as an underwound helix as compared to the canonical B-DNA structure [36,37]. This finding was supported by crystal structures of the PCL proteins MTF2 and PHF1 bound to DNA, demonstrating a binding of the winged-helix domain to DNA with unmethylated CpGs and reduced helical twist [37]. The loss of MTF2 in mESCs resulted in globally reduced H3K27me3 levels, and recruitment of EZH2 was also severely reduced at target gene locations [36,45,46,62]. Unlike MTF2, the loss of other PCL proteins, PHF1 and PHF19 did not elicit a similar level of drastic reduction of PRC2 recruitment in mESCs [45,62]. This might be because these proteins are expressed at much lower levels compared to MTF2 in mESCs [63].

The other PRC2.1 protein that plays an important role in mESCs is EPOP. Deletion of EPOP in mESCs prevents PRC2.1 from interacting with Elongin B/C proteins, which in turn reduces RNA polymerase II elongation on target genes [64]. Therefore, while the loss of EPOP does not directly translate into loss of H3K27me3 levels, it would set up for a low level of permissive transcription at target genes and affect how PRC2 regulates transcription. The interaction of EPOP with PRC2 is found to be mutually exclusive with PALI1/2 [38,40]. This is corroborated by studies demonstrating a dynamic shift of the PRC2.1 interactome between EPOP and PALI1/2 during mESC differentiation [63]. Unlike EPOP, the ablation of PALI1/2 proteins resulted in some degree of H3K27me3 reduction at target genes [39]. This indicates the importance of these proteins in maintaining the catalytic function of PRC2, but just like PHF1 and PHF19, their stoichiometry relative to other PRC2.1 proteins is found to be minimal.

### 3.2. PRC1 Variant Complexes and Their Roles in Embryonic Stem Cells

Canonical PRC1 complexes contain CBX proteins (CBX2, CBX4, CBX6, CBX7, or CBX8) (Figure 1). The CBX proteins exhibit different patterns of expression, with CBX7 being highly abundant in undifferentiated stem cells. While the CBX proteins can bind to H3K27me3 (deposited by PRC2) via their chromodomains, some single-molecule quantification studies demonstrated that only CBX7 and CBX8 proteins were displaced from chromatin in PRC2 knock-out mutant mESC lines [65], revealing that other mechanisms might be in place, such as CBX2 inducing phase separation for PRC1, to regulate gene expression [66,67]. Aside from the CBX proteins, the PCGF proteins that are found in cPRC1—PCGF2 (MEL18) and PCGF4 (BMI1), when genetically removed in mESCs appear to have little to minimal effect on global levels of H2AK119ub and transcription, despite showing a strong decrease in PRC1 subunit binding to chromatin [27,28,68,69]. This indicates that the regulation of H2AK119ub levels is largely independent of cPRC1 these variant complexes, at least in the mESCs.

In ncPRC1, components such as KDM2B, part of the PRC1—PCGF1 variant complex, have been demonstrated to be crucial in the recruitment of RING1B to unmethylated CG dense regions via their zinc finger domains [70–72]. In PRC1—PCGF3/5 complexes, both PCGF3 and PCGF5, when deleted together, result in a decrease in H2AK119ub levels and in mESCs. Both proteins were shown to be able to recruit RING1B to chromatin [28].

## 4. Roles of Polycomb Complexes in Developmental Cell Lineages

The repertoire of different phenotypes observed for many mutants of different Polycomb proteins, and their relatively delayed phenotypes suggest that some of these variant

complex subunits might play more important roles during organogenesis and later developmental stages (Figure 3). Recent studies looking into the roles of different Polycomb proteins during in vitro differentiation shed some light on the potential specific mechanisms and roles of these different proteins during development. Figure 4 provides an overview and developmental roadmap of the known contributions of Polycomb complexes during development. It is apparent that there are still many developmental cell types, particularly from endoderm lineages, that remain to be investigated for contributions of Polycomb complexes.

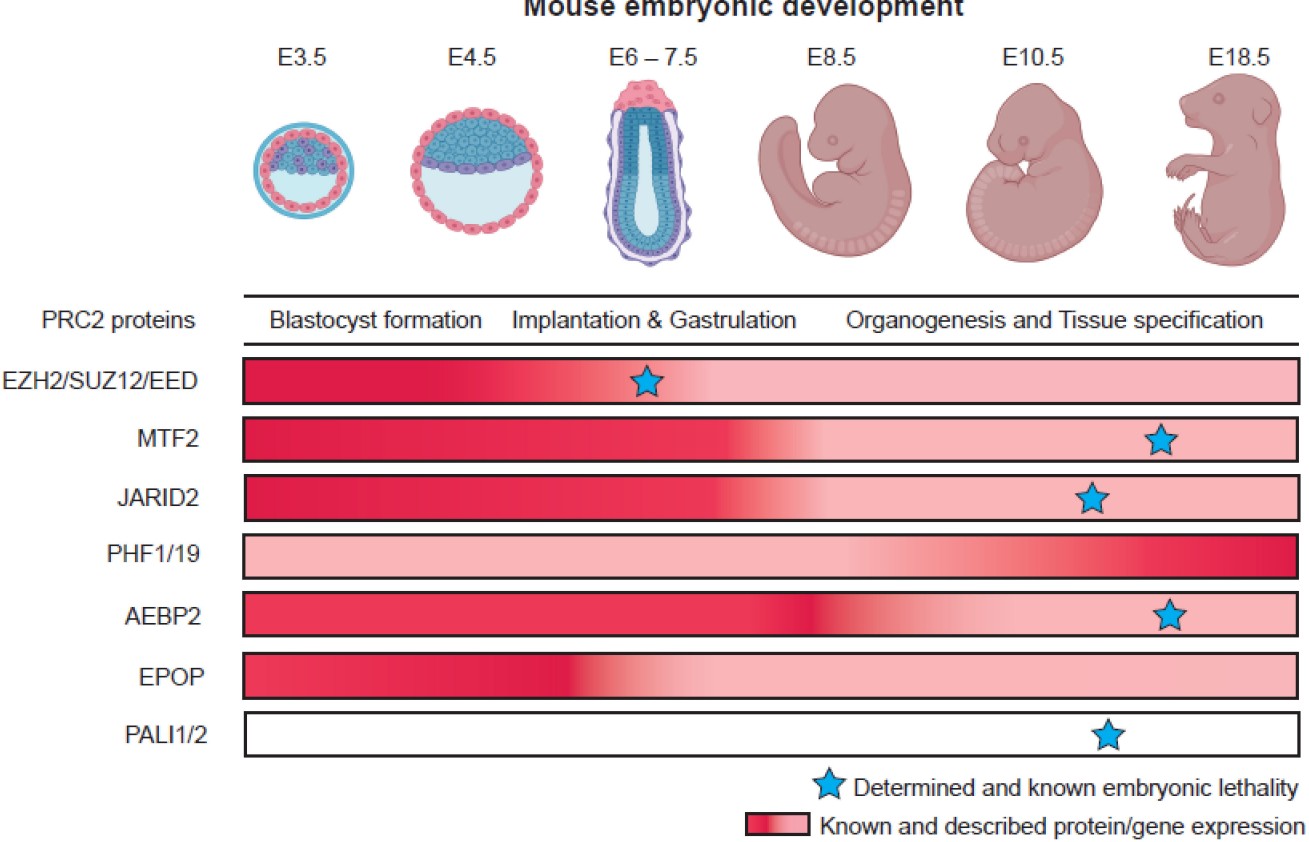

**Figure 3.** Differences in pattern of Polycomb group protein expression and their embryonic loss-of-function phenotypes. Embryo pictures reprinted from "Mouse Development", by BioRender.com (accessed on 15 July 2022). (2022). Retrieved from https://app.biorender.com/biorender-templates (accessed on 15 July 2022).

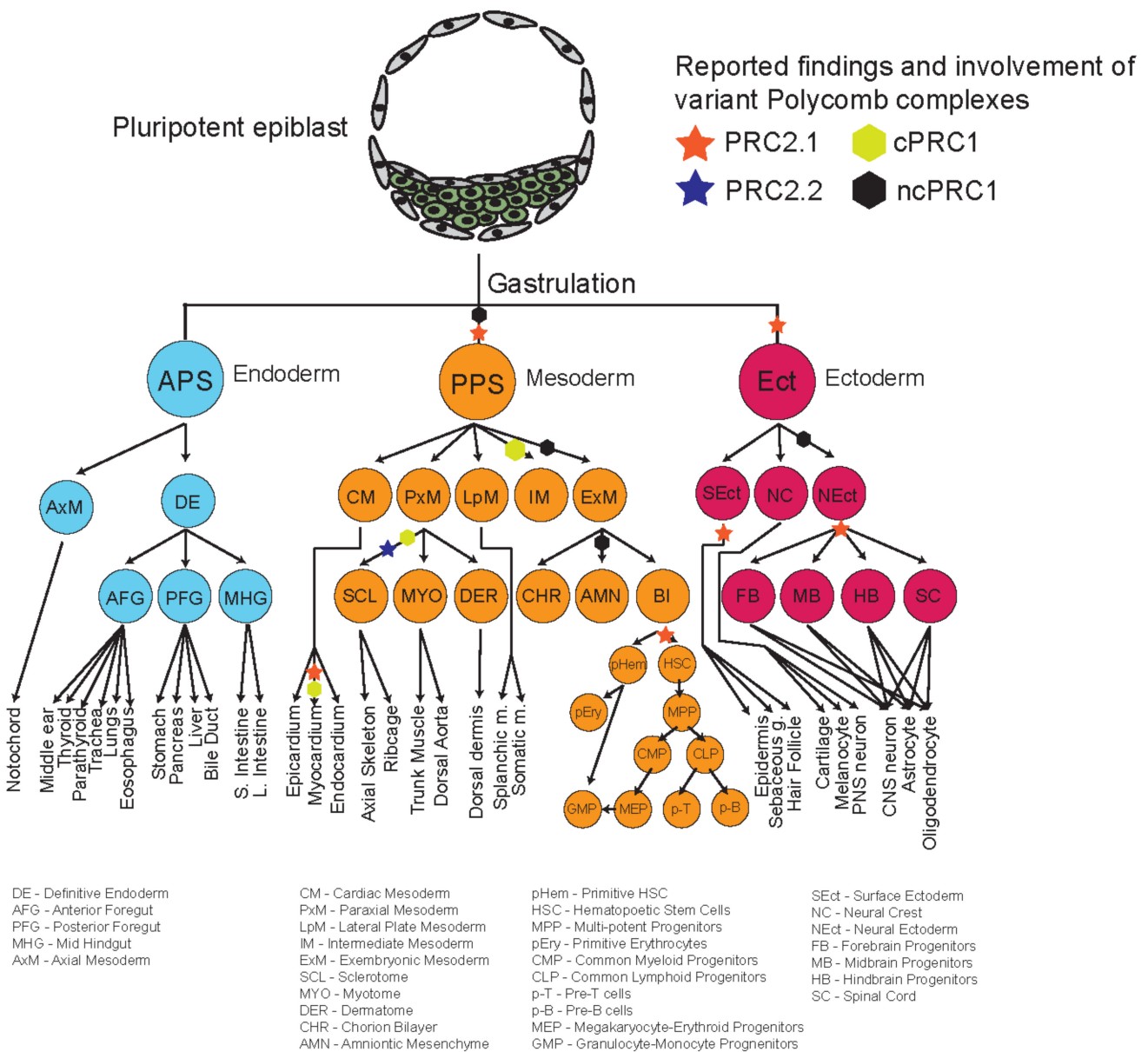

**Figure 4.** Developmental roadmap charting Polycomb function and involvement at different key lineage decision steps.

### 4.1. Dynamic Functions of PRC2 Variant Complexes

In PRC2.1, the PCLs are known for their dynamic expression pattern during differentiation (Figure 3). In mouse embryonic stem cells (ESCs), MTF2 expression is high and is much more abundant in the PRC2 complex compared to the other PCLs, PHF1 and PHF19 [63]. This changes during differentiation, when PHF1 and PHF19 increase in abundance while the levels of MTF2 decrease [40,63,73]. It is noteworthy that all three PCLs contain one Tudor domain, two PHD fingers, an extended homology domain, and a conserved C-terminal domain. Despite their similarities, PHF1 and PHF19 can associate with transcriptional elongation mark H3K36me3 through their Tudor domain whereas MTF2 cannot [74–77]; this histone modification inhibits PRC2 activity on target genes. Both PHF1 and PHF19 modulate the recruitment of PRC2 to target genes previously marked by H3K36me3 via their Tudor domains. Furthermore, it has been shown that PHF19 could recruit NO66, which is a H3K36me3 demethylase, to remove the transcriptionally active mark and promote deposition of H3K27me3 during differentiation [78].

Though the early-to-late shift in PCL expression during differentiation (MTF2 most abundant early, PHF1 and PHF19 more abundant later) may explain some of the phenotypes observed in loss-of-function studies, this may be a too simple scheme to conceptualize all findings. Their roles are not absolute, may differ in nature and extent between cell lineages, and may overlap in developmental time. Single-cell transcriptomic analyses of *Mtf2* null embryoid bodies revealed that the loss of MTF2 allowed for a much more rapid differentiation towards all germ layers and lowered the epigenetic threshold for the exit of pluripotency [8,79]. In addition, both MTF2 and JARID2 are required for normal differentiation of neural progenitor cells [79]. Similarly, hematopoietic progenitors of MTF2 null mice were unable to properly differentiate into mature blood cells, resulting in severe anemia [18]. PHF1 and PHF19, on the other hand, were found to have increased interactions with core PRC2 complexes during in vitro differentiation towards neural precursors [63]. Recent findings in myeloid cells also suggest an important role for PHF19 in regulating erythroid differentiation by targeting a cell cycle inhibitor [80]. PALI1/2 null embryos were also able to survive until prenatal stage despite having a reduction of H3K27me3 levels [39]. Often, important signaling modulators (for example from the WNT and FGF pathways) and lineage-specific transcription factors (for example GATA2, IRX3) are, when already poised for expression, derepressed in accessory subunit mutants [8,18], explaining the developmental context in which the phenotypic defects are observed.

Unlike PRC2.1, *Jarid2* (PRC2.2) null mutants displayed a predominant differentiation towards early differentiating precursors and reduced efficiency towards mesendodermal lineages [8]. *Jarid2* null mouse embryos were also unable to develop beyond E11.5 and have defects in heart formation and neural tube closure [61]. AEBP2 null mice seems to elicit a strong phenotype later during development, exhibiting anterior homeotic transformation and pre-weaning lethality [22,23].

Together, the phenotypic differences observed in different Polycomb mutant reflect two points. First, the delay in developmental phenotypes observed in PRC2.1 and PRC2.2 accessory subunit mutants compared to core mutants highlights a potential redundancy between the complexes. Losing one variant complex does not remove the other, and the phenotypic outcomes might be a result of partial loss of activity of PRC2 and skewed effects on subsets of PRC2 target genes, as PRC2.1 and PRC2.2 are targeted to genes in different ways. Next, the variety of late-stage phenotypes suggests potential lineage specific roles of different variant complexes. This could be driven by tissue specific expression of variant complexes to refine and maintain lineage specificity rather than regulating early pluripotency exit network.

### 4.2. Dynamic Functions of PRC1 Variant Complexes

Like PRC2, PRC1 variant complexes also demonstrate varying degrees of phenotypes in different tissues and developmental time points (Table 1 and Figure 4).

The different CBX proteins of the cPRC1 variant complex each display a different phenotype upon their loss-of-function. *Cbx2* null mice were able to survive embryonic development, albeit dying within few weeks of birth [81]. They also present with small body masses and alterations in sex organ development. Studies which showed the role of CBX2 in repression of ovary-determining genes further enhanced the specific role of this cPRC1 variant in sex organ lineages [81]. *Cbx4* mutants also demonstrated a later lethal phenotype, with mice dying before weaning, mainly due to growth retardation [82]. Overexpression of CBX4 protein was able to attenuate the development of osteoarthritis in mice and suggest a specific role of this PRC1 variant in during later stages of development [83]. *Cbx7*-depleted mESCs gave rise to ectodermal enriched teratomas, which points towards a more ectodermal commitment role for this variant of PRC1 [82,84]. Other CBX proteins such as CBX6 and CBX8 do not demonstrate any severe lethality apart from some organ defects in *Cbx6* mutant embryos [82]. Apart from CBX proteins, the PHC proteins, another component of the cPRC1 variant complex, also display different sets of phenotypes. The role of PHC2 seems to be important in proper skeletal formation and *Phc3* null mice

develop severe cardiac abnormalities [85]. PCGF2 and PCGF4 are also part of the cPRC1 complex, and it has been shown that the latter is important during hematopoiesis and axial skeleton transformation stages during development, particularly during Hox gene expression regulation [86,87]. These developmental defects are phenocopied by the loss of PCGF2 protein, which compensates the activity of PCGF4 [88]. Both core components of the ncPRC1 complex, RYBP and YAF2, are critical in early mouse embryonic development, as mutant mice were unable to survive past gastrulation [89]. This demonstrates the relative importance of ncPRC1 in early stages of development as compared to cPRC1 variants. The deletion of ncPRC1 component Kdm2b results in embryonic lethality at mid-gestation and Pcgf1 null mice were unable to survive past E12.5 [82,90]. Pcgf1 null mESCs also display defects in ectoderm and mesoderm specification, accompanied by a deregulation of key transcription factors required for those lineages [91]. Contrastingly, AUTS2, a component of ncPRC complex, is implicated in brain development with additional defects in the mesendodermal germ layer lineages, such as the heart and gut [92]. Finally, Cbx3 null mice showed defects in primordial germ cell differentiation and exhibited severe impairment in spermatogenesis [93,94].

## 5. Towards a Systematic Analysis of Polycomb Function in Development

The data reviewed above establish the relative importance of different subunits and Polycomb complexes in various tissues and stages of development. This relative importance is contextualized by expression levels of all components (Figure 3), the buffering capacity in the Polycomb system and the identity and nature of transcriptional activators antagonized by the Polycomb system (Figure 2). We can say with strong certainty that different variants or 'flavors' of Polycomb, be it PRC1 or PRC2, have important roles in specific stages of development. With the large variety of different phenotypes and potentially varying mechanisms of Polycomb regulation presented, we believe it is time to study these different components in a systematic manner to reconciliate the findings and compile a roadmap filled with the roles of different Polycomb proteins throughout development. To do so, we must establish robust systems and incorporate sensitive modern technologies to dissect the heterogeneity of an inherently complicated process—cellular differentiation. Recent advancements in cellular differentiation allow us to harness the power of stem cell differentiation to establish a system to study specific roles of Polycomb proteins in developmental processes that model embryonic development. This is complementary to in vivo studies, that, often are more limited in mechanistic resolution due to the complexity of the embryo and the restrictions imposed by intra-uterine development.

Systematic interrogation of the roles of different and any Polycomb mutants in a fast, sensitive, and robust manner is now possible (Figure 5). It is crucial that different Polycomb mutant cell lines are generated using CRISPR-mediated knock-out technology in genetically matched mouse cell lines to reduce technical bias. Exquisite additional insight can be generated by using rapid, inducible, and reversible mutant lines with endogenous knock-in of an Auxin-inducible degron (AID) tag to the gene-of-interest. This allows for powerful switches to control the expression of Polycomb proteins at any time during differentiation experiments [95,96].

| Polycomb mutant cell lines<br>+<br>Genetically matched WT line | Grow cells in suspension cultures<br>-<br>3D Embryoid Bodies or Gastruloids | Spontaneous differentiation<br>+<br>Single-Cell RNA, ChIP, ATACseq |
| :---: | :---: | :---: |
| 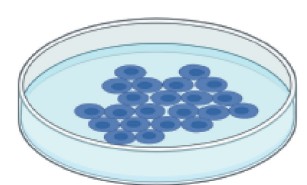 | 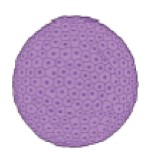 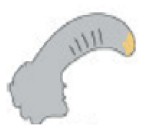 | 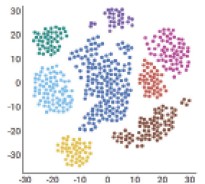 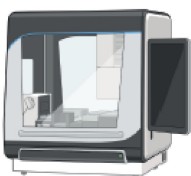 |

(1) Gather which cell types are generated by which mutants

(2) Infer the difference in differentiation speeds between mutants

(3) Analyze the molecular mechanisms driving the differences at
RNA and histone modifications level

**Figure 5.** Modern approaches and technological advances in studying Polycomb functional biology. With illustrations reprinted from "Petri dish with cells", "3D organoids", "Single-cell RNA seq cluster graph", and "10X Genomics" by BioRender.com (accessed on 15 July 2022). (2022). Retrieved from https://app.biorender.com/biorender-templates (accessed on 15 July 2022).

### 5.1. In Vitro 2D and 3D Culturing Systems to Study Polycomb Functions

One of the best ways to study the role of Polycomb proteins during differentiation is to spontaneously differentiate Polycomb mutant mESCs as aggregates of pluripotent cells (Embryoid Bodies, EBs). The EBs allow for better mimicking of in vivo conditions by permitting direct cell–cell interactions, signaling, and morphogenetic changes that usually occur during development. The conditions of EBs culture should be consistent, particularly the sizes of EBs during differentiation should be controlled (for example, starting with ~200 cells per EB at the start of the aggregation). To properly untangle the effects of the loss of the Polycomb gene during differentiation, these experiments should be performed with precise conditions replicated for each mutant cell line. The development of gastruloid [97] and blastoid [98] 3D cultures, which resemble aspects of respectively gastrulation and blastocyst development, enables investigations into mechanisms targeting specific timepoints of development.

Apart from 3D systems, current approaches in monolayer (2D) differentiation systems are also highly efficient and suitable for studying Polycomb biology. Many directed differentiation protocols make use of in vivo signaling dynamics to direct the differentiation of mESCs towards pure populations of desired cell types in a short amount of time. This is particularly useful to validate the observations and results from the 3D cultures, which are usually more heterogenous in nature. Differentiation in monolayer culture also allows for robust testing of signaling pathways during key differentiation stages.

### 5.2. Untangling the Functions of Polycomb Proteins Using Single-Cell Omics Technologies

As mentioned above, differentiation of 3D cultures is a highly heterogenous process. One of the best ways to tackle this issue is by performing single-cell omics on the dissociated 3D cultures. It is currently possible to extract the RNA, chromatin accessibility, histone modifications, and proteomic information of individual cells, not only separately, but also multi-modally in the same cells [99]. These techniques allow us unprecedented access into the transcriptome and epigenome of single cells in a heterogeneous population of differentiated cell types. After proper annotation of every single cell in the dataset, it will be possible to reveal specific differences in EBs of mutants versus wild type cells. Using this approach, inferences about speeds and biases of differentiation in different cell lineages

can be made, while also allowing a careful dissection of the molecular mechanisms which drive the differences between mutant cell lines.

## 6. Concluding Remarks

In this review, we have highlighted the numerous phenotypes of different Polycomb proteins during development. Different Polycomb complexes play specific and separate roles in different timepoints or lineages during development. While the core components are mostly important in early embryonic processes, many other variant subunits also have crucial roles, for example in late organogenesis, sex determination, and body axis formation. The variety of these functions played by different complexes highlight the complex yet fascinating mechanisms of Polycomb proteins throughout development. Recent progress in the field has made new inroads in both our molecular and our developmental understanding of Polycomb function. However, our developmental roadmap of Polycomb function (Figure 4) demonstrates clearly that several lineages are still understudied when it comes to Polycomb regulation, such as cell types from the endoderm lineage. It is unknown if this is due to reporting bias or if this is simply a reflection of the interest of the field in different tissues and cell lineages. It would be interesting to further define the 'flavors' of Polycomb complexes, not only in late-stage development but also mature adult tissues. Since Polycomb biology is heavily implicated in diseases such as cancer, it is crucial that we fill in the gaps of Polycomb biology to fully understand how gene repression goes awry in disease and devise therapeutic interventions accordingly.

**Author Contributions:** C.H.L. conceptualized, wrote and edited the manuscript. G.J.C.V. conceptualized, reviewed and edited the manuscript. All authors have read and agreed to the published version of the manuscript.

**Funding:** This research is supported by Radboud University.

**Institutional Review Board Statement:** Not applicable.

**Informed Consent Statement:** Not applicable.

**Data Availability Statement:** Not applicable.

**Conflicts of Interest:** The authors declare no conflict of interest.

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
