# Peer review of "The Role of Polycomb Proteins in Cell Lineage Commitment and Embryonic Development"

_2075-4655, 2022_

Round 1

Reviewer 1 Report

This manuscript describes important roles of polycomb proteins with focused on its specific complexes. The roles of polycomb proteins are mentioned on the various timelines, which would be comprehensible to readers. Polycomb proteins, which has a complicated function, are generally difficult to grasp the whole aspect, hence it is important to arrange the relations between specific functions and protein complexes.

Although the manuscript is well prepared, there are some typographic errors, which should be modified before the resubmission of a revised version.

 For example; 1) page 3, line 117, add period after ‘activity’. 

2) page 4, line 202, remove the space (just before MEL18). 

3) page 5, line 250, add period after ‘H3K27me3 levels’.

Author Response

Response to reviewers

Author responses are preceded by “>>>”.

>>> We thank both reviewers for their positive and constructive feedback on our review which helped to make it even better. We have corrected the typographic errors that were pointed out by reviewer 1. We also gratefully used the comments of reviewer 2 to improve the text for conceptual clarity and structure. The changes made in the revised text can be found using the ‘Track Changes’ function in MS Word.

Specific points

Reviewer 1

This manuscript describes important roles of polycomb proteins with focused on its specific complexes. The roles of polycomb proteins are mentioned on the various timelines, which would be comprehensible to readers. Polycomb proteins, which has a complicated function, are generally difficult to grasp the whole aspect, hence it is important to arrange the relations between specific functions and protein complexes. Although the manuscript is well prepared, there are some typographic errors, which should be modified before the resubmission of a revised version.

For example; 1) page 3, line 117, add period after ‘activity’.

>>> Changed accordingly.

2) page 4, line 202, remove the space (just before MEL18).

>>> Changed accordingly.

3) page 5, line 250, add period after ‘H3K27me3 levels’.

>>> Changed accordingly.

Reviewer 2 Report

I preface this report by saying the version supplied for review did not contain the figures.

I cannot provide any guidance on whether they are appropriate, helpful, or not. This is not a reflection on the authors, but on the publisher.

The role of Polycomb proteins in cell lineage commitment and embryonic development

This review by Loh and Veenstra describes the landscape of polycomb complexes focusing on their composition and mutant phenotypes. The descriptions of core versus variant subunits for each complex is comprehensive and useful to a general readership. Likewise cataloguing the phenotypes of null mutations for core versus variant subunits paints an important picture about different stages of development and how refinement of cell type is paralleled by diversification of polycomb proteins required for normal development.

Overall, I learned a lot by reading this review.

I have some suggestions detailed below that may help the authors in improving the ‘readability’ of their manuscript which was frankly challenging in some places.

Specific comments for authors:

Section 1. Introduction nicely sets up the differences between PRC1 and PRC2 and provides a cogent description of each complex as being composed of a common core set of subunits found in complex with variant subunits. The differential phenotypes of null mutations in core subunits (catastrophic) versus variant subunits (variable) is likewise nicely described.

Section 2. molecular functions

Nice description of PRC1 complexes and definition of the key differences between the canonical and non-canonical forms emphasizing both subunit differences (and ability to recognize H3K27me) and differences in catalytic properties.

Description of PRC2 seems a little non-linear to me, particularly the description of PRC2.1 (lines 91-98).

Sentence in line 105 needs work – catalysis..can be bound? Maybe the catalytic product? In fact this entire paragraph (lines 105 – 116) seems to me a listing of observations with little intellectual connection. Perhaps the figure would have helped me here…

The following paragraph (lines 118 – 138) is well written, logical and clear.

Section 3 pluripotency and development

I do not entirely agree with the assertion here (line 155) that later developmental phenotypes of variant polycomb group proteins are less severe than those of core subunits – these embryos are also dead, right? Timing of an essential function does not (my opinion) indicate severity or lack thereof.

Typo in line 205 – largely independent of cPRC1 these variant. Probably should read – largely independent of these variant cPRC1…

Section 4 developmental cell lineages

Opening paragraph of this section (lines 213 – 224) could certainly be improved. I am unclear on what the authors are attempting to convey in sentence 2 – lines 216-218. Last sentence of the paragraph (lines 222 – 224) is awkward.

First paragraph of Dynamic functions of PRC2 variant complexes (lines 225 – 252) appears to be a listing of seemingly unconnected findings. Perhaps some level of synthesis of these findings by the authors into a coherent conclusion would be useful?

Section 5 systematic view

The discussion of how to tackle the complicated regulation by polycomb complexes is useful and well organized – replete with useful suggestions for future work.

Author Response

Response to reviewers

Author responses are preceded by “>>>”.

>>> We thank both reviewers for their positive and constructive feedback on our review which helped to make it even better. We have corrected the typographic errors that were pointed out by reviewer 1. We also gratefully used the comments of reviewer 2 to improve the text for conceptual clarity and structure. The changes made in the revised text can be found using the ‘Track Changes’ function in MS Word.

Specific points

Reviewer 2

Section 2. molecular functions

Nice description of PRC1 complexes and definition of the key differences between the canonical and non-canonical forms emphasizing both subunit differences (and ability to recognize H3K27me) and differences in catalytic properties.

Description of PRC2 seems a little non-linear to me, particularly the description of PRC2.1 (lines 91-98). Sentence in line 105 needs work – catalysis..can be bound? Maybe the catalytic product? In fact this entire paragraph (lines 105 – 116) seems to me a listing of observations with little intellectual connection. Perhaps the figure would have helped me here…

>>> We removed a redundant sentence that made the paragraph non-linear and have edited the paragraph for consistency and clarity. For the reader it will also help to have Figure 1 that provides an overview of the Polycomb complexes.

The following paragraph (lines 118 – 138) is well written, logical and clear.

Section 3 pluripotency and development

I do not entirely agree with the assertion here (line 155) that later developmental phenotypes of variant polycomb group proteins are less severe than those of core subunits – these embryos are also dead, right? Timing of an essential function does not (my opinion) indicate severity or lack thereof.

>>> The reviewer makes an excellent point and we agree that discussing the roles of PRC2 subunits in terms of importance was not the best choice of words given that they are all essential for viability. For this review, it is still important to reflect on what the differences in phenotypes and the stages they occur may mean. We exchanged the word “important” for “subtle”, to reflect the reality that a complete loss of a certain activity (mediated, for example by core subunits of PRC2) may affect an embryo at an earlier stage than a partial loss of activity, which would allow the embryo to “get by” for a while. In addition to this notion, we also expanded on other aspects that contextualize PRC2 function and that may be relevant for the differences in phenotypes. 

Typo in line 205 – largely independent of cPRC1 these variant. Probably should read – largely independent of these variant cPRC1…

>>> Changed accordingly

Section 4 developmental cell lineages

Opening paragraph of this section (lines 213 – 224) could certainly be improved. I am unclear on what the authors are attempting to convey in sentence 2 – lines 216-218. Last sentence of the paragraph (lines 222 – 224) is awkward.

>>> We deleted the second and rephrased the last sentence of the paragraph.

First paragraph of Dynamic functions of PRC2 variant complexes (lines 225 – 252) appears to be a listing of seemingly unconnected findings. Perhaps some level of synthesis of these findings by the authors into a coherent conclusion would be useful?

>>> The reviewer is right to note that not all observations can be put in a simple conceptual frame. At the beginning of the paragraph we discuss this, including why not all observations may fit in a simple scheme. We also split this lengthy paragraph so it becomes more obvious that in the first paragraph we are talking about the mechanistic differences of PRC2.1 complexes containing MTF2, PHF1 and PHF19 with their expression dynamics during differentiation, and paragraph 2 focusing on the functional observations from the loss of those proteins.

Section 5 systematic view

The discussion of how to tackle the complicated regulation by polycomb complexes is useful and well organized – replete with useful suggestions for future work.